# Graphene Infused Ecological Polymer Composites for Electromagnetic Interference Shielding and Heat Management Applications

**DOI:** 10.3390/ma14112856

**Published:** 2021-05-26

**Authors:** Klaudia Zeranska-Chudek, Anna Wróblewska, Sebastian Kowalczyk, Andrzej Plichta, Mariusz Zdrojek

**Affiliations:** 1Faculty of Physics, Warsaw University of Technology, Koszykowa 75, 00-662 Warsaw, Poland; anna.wroblewska@pw.edu.pl (A.W.); mariusz.zdrojek@pw.edu.pl (M.Z.); 2Faculty of Chemistry, Warsaw University of Technology, Noakowskiego 3, 00-664 Warsaw, Poland; skowalczyk@ch.pw.edu.pl (S.K.); andrzej.plichta@pw.edu.pl (A.P.)

**Keywords:** graphene, polymer composite, dual-functional material, microwave X-band, electromagnetic shielding, thermal conductivity

## Abstract

In the age of mobile electronics and increased aerospace interest, multifunctional materials such as the polymer composites reported here are interesting alternatives to conventional materials, offering reduced cost and size of an electrical device packaging. We report a detailed study of an ecological and dual-functional polymer composite for electromagnetic interference (EMI) shielding and heat management applications. We studied a series of polylactic acid/graphene nanoplatelet composites with six graphene nanoplatelet loadings, up to 15 wt%, and three different flake lateral sizes (0.2, 5 and 25 μm). The multifunctionality of the composites is realized via high EMI shielding efficiency exceeding 40 dB per 1 mm thick sample and thermal conductivity of 1.72 W/mK at 15 wt% nanofiller loading. The EMI shielding efficiency measurements were conducted in the microwave range between 0.2 to 12 GHz, consisting of the highly relevant X-band (8–12 GHz). Additionally, we investigate the influence of the nanofiller lateral size on the studied physical properties to optimize the studied functionalities per given nanofiller loading.

## 1. Introduction

Multifunctional materials combine two or more functionalities such as durability, self-healing, or electrical conductivity within one material [1]. Out of several types of multifunctional materials, we will concentrate on polymer composites with nanoscale fillers, also called multiscale composites or nanocomposites. Polymer composites join the mechanical sturdiness of the polymer matrix with unique traits of chosen nanofillers. Polymer composites offer a plethora of advantages over conventional materials, such as high durability, resistance to corrosion, low weight, or the possibility of using well-known and large-scale production methods developed for polymers, e.g., 3D printing, extrusion, or injection molding. 

In present-day electronics, an electronic device’s packaging usually has to provide efficient shielding of stray electromagnetic interference and efficient heat management. The minimum requirement for industrial applications for EMI shielding efficiency (EMI SE) is 10 dB [2], which corresponds to shielding 90% of incident radiation. A similar requirement for thermal conductivity is set to 1 W/mK. Conventionally those two functionalities are realized via two different materials—an EMI shield and a thermal interface material (TIM). Combining those two functionalities in one material, yielding a so-called dual-functional material, could significantly reduce the cost and weight of a device packaging and reduce the complexity of a packaging design. That dual-functional approach has another asset regarding EMI shielding materials, whose main shielding mechanism is absorption. The energy absorbed during the shielding process is often transformed into thermal energy, and thus, especially in high-energy applications, good heat management could prevent overheating of the protected device. 

The concept of such dual-functional polymer composites is relatively new. However, it has been gained the interest of other scientific groups. Kargar et al. report epoxy polymer composite filled with high loading (up to 50 wt%) of few-layer graphene flakes [3]. The reported material exhibits 45 dB EMI shielding efficiency in the microwave X-band and thermal conductivity of 8 W/mK. Similarly, Lee et al. show a metal-based polymer composite with extremely high EMI SE and thermal conductivity [4]. Both high EMI SE and thermal conductivity stem from the conductive nanofillers. From all available nanofillers, graphene nanoflakes seem to fit perfectly in the dual-functional requirements. Graphene flakes are highly conductive [5,6], absorb EM radiation in a broad range [7,8,9], and are capable of good heat management [10,11,12]. Several scientific works explore graphene flakes and materials based on them in light of one of the two mentioned functionalities [13,14,15,16,17,18]. The properties of graphene flakes and their influence on a polymer composite can change with their loading and geometry. Both the thermal and EMI shielding properties of graphene-based materials seem highly influenced by the graphene loading and the formation of a conductive network within the polymer matrix [16,19,20]. In the EMI shielding theory, it has been postulated that the absorption coefficient depends on the specific geometry of a nanofiller—particles with a higher aspect ratio yield higher absorption of EM radiation [21]. As there is usually a cap for nanofiller content dispersed in a particular polymer, investigation of the influence of particle geometry on EMI shielding may help maximize the desired functionalities per specific nanofiller loading. 

Here we present a study on dual-functional po(lylactic acid) (PLA) polymer composites imbued with graphene nanoplatelets (GNP). The dual-functionality is expressed in high—over 40 dB EMI SE in a broad microwave range between 2 and 12 GHz (including the important microwave X-band) and thermal conductivity of ~1.7 W/mK. We show how EMI SE, thermal and electrical conductivity, and absorption coefficient evolve with changing nanofiller loading of up to 15 wt%. We also analyze the influence of the filler aspect ratio on the aforementioned properties by studying composites infused with three GNP lateral sizes—0.2, 5, and 25 μm—which has not been discussed in this frequency range. We distinguish the reflection and absorption components of EMI SE and specify absorption as the main shielding mechanism in polylactic/graphene nanoplatelet composites.

## 2. Methods and Materials

### 2.1. Material: PLA/GNP Composites

For this work, three series of polymer composites with nanocarbon inclusions were fabricated. The PLA was infused with GNP of three different lateral sizes—0.2, 5, and 25 um, referred to from now on as PLA/GNP composites. The PLA was purchased from Resinex in the form of a powder. The GNP were provided by Sigma Aldrich (XG Sciences, Inc., Lansing, MI, USA) also in the form of a powder. The average thickness of a GNP was 15 nm which corresponds to approximately 40 layers of graphene.

The polymer composites were fabricated as follows. The PLA and GNP were mixed in a lab mixer in specified proportions to receive mixes with 0.5, 1, 2, 5, 10, and 15 wt% (weight percent) of graphene in the mixture. Each sample was produced in series of three to ensure better statistical averaging of the data points. Next, the mixtures were hot-pressed on a hydraulic press equipped with two heated plates and a mold in the shape of a 3 × 3 × 0.1 cm^3^ cuboid and tensile strength test species in the shape of “dog bone” of the length of 8 cm in total and 3.5 cm in the testing area, the thickness of 0.2 cm and width of 0.5 cm in the testing area. The hydraulic press was heated up to 200 °C, and a pressure of 2 MPa was applied for 15 s. Next, a degassing step was conducted. Finally, the proper press was executed in the pressure of 7 MPa for 45 or 60 s for cuboids and “dog bones”, respectively. Hot press equipment used for the sample fabrication was Carl Zeiss, DP 36 (Oberkochen, Germany).

### 2.2. Characterization Methods

Standard characterization was conducted to assure the quality of the fabricated material. Figure 1a shows a picture of PLA/GNP composite and a pure PLA reference sample. The opacity change in the GNP-loaded composite is visible. SEM pictures were taken using Raith E-line+ (Dortmund, Germany) to check the homogeneity of the dispersion of graphene flakes in a polymer matrix. Figure 1b,c show SEM pictures of a PLA/GNP composite infused with 25 μm GNP at 15 wt% loading. The SEM pictures show the distribution of the GNP in the polymer matrix and provide a rough estimate of GNP lateral size. They confirm the random flake distribution and consistency of GNP size with the datasheet given by the GNP provider.

Raman spectra were collected using a laser excitation line (λ = 532 nm) with a back scattering configuration in Renishaw inVia spectrometer (UK). As seen in Figure 2a, which shows Raman spectra collected for the composites with the smallest size of graphene flakes (0.2 μm), we observed polymer bands (~1454, 1771 cm^−1^ and three peaks between 2800 and 3000 cm^−1^ [22]) and bands characteristic for graphene (G and 2D around 1580 cm^−1^ and 2700 cm^−1^ respectively). The graphene characteristic bands were not observed in pure polymer samples. Analogically we have observed the decline of PLA characteristic bands for samples with high GNP loading. See Appendix A for the comparison of Raman spectra for different GNP loadings and Raman spectra of composites with different GNP lateral sizes.

Raman spectroscopy was used to investigate the dispersion of carbon nanofillers within the polymer matrix. Raman maps (100 × 100 μm^2^, containing 676 spectra, collected in focus-track mode) and visible in Figure 2 depict the dispersion of GNP in PLA/GNP composites with different GNP loading. Figure 2b shows the G band position (white color indicates lack of G band in the spectrum) for a sample containing 0.5 wt% graphene flakes in the polymer matrix. One can see that the characteristic graphene bands are present only in a few places on the sample surface, which indicates a non-uniform distribution of GNP in PLA or the GNP fraction is too small to show on the Raman spectra. The same analysis was conducted for a sample containing 15 wt% graphene flakes (Figure 2c), which showed a more uniform GNP distribution via the presence of characteristic bands over the entire investigated area.

Tensile strength tests were conducted by the Instron 5566 Universal Testing Machine (Norwood, MN, USA), equipped with a 10 kN measuring head and self-tightening roller tensile grips. Tensile strength tests were performed at room temperature at a running rate of 20 mm/min. Test species in the shape of “dog bone” comprising 10 wt% of graphene in the mixture with PLA were used. Data were processed with BlueHill2 software (Instron, Norwood, MN, USA). The obtained results for composites containing GNP of various flake sizes: 0.2, 5, and 25 μm are shown in Figure 3. The introduction of 10 wt% of any GNP to neat PLA causes a decrease in tensile strength and maximum elongation and an increase in material stiffness. However, the tensile strength of obtained composites is higher than for neat aliphatic polyolefins, e.g., HDPE [23,24,25].

Finally, we have conducted measurements needed to investigate the assumed dual-functionality of PLA/GNP composites. Conductivity σ was calculated based on volume resistivity ρ. The volume resistivity was measured using a QWED single post dielectric resonator (Warsaw, Poland) with an operating frequency of 5 GHz. Thermal constants such as thermal conductivity, thermal diffusivity, and specific heat capacity were procured using a Hot Disk Thermal Constants Analyzer TPS 2500 S (Gothenburg, Sweden). The measurements were conducted using a Hot Disk 5501 Kapton Sensor (radius 6.403 mm), 20 mW power for the 40 s in ambient temperature and atmosphere.

EMI shielding efficiency in the microwave range was studied via a setup consisting of an Agilent Vector Analyzer N5221A PNA (Santa Clara, CA, USA) and a custom coaxial sample holder made in compliance with an ASTM D4935-10 Standard. This setup enables measurement of the scattering matrix components (reflection components S_11_ and S_22_, as well as transmittance components S_12_ and S_21_). The measurements are conducted in a broad microwave range—from 0.2 to 12 GHz.

The elements of the scattering matrix (S_11_, S_12_, S_21,_ and S_22_) received by comparing the signal from two ports of a vector analyzer—port 1 and 2, can be linked to transmittance and reflectance in the following way: |S_21_|^2^ = |S_12_|^2^ = T and |S_11_|^2^ = |S_22_|^2^ = R, where T is the transmittance and R is the reflectance. T and R can, in turn, be used to calculate the EMI shielding efficiency and its components. The entire amount of shielded radiation is called the total shielding efficiency (SE_TOT_). EMI shielding efficiency is usually described with decibels (dB), EMI SE of 10 dB can be translated to shielding of 90% of incident radiation. The SE_TOT_ can be divided into three components—reflection (SE_R_), absorption (SE_ABS_), and multiple internal reflections (SE_MIR_), in a way that SE_TOT_ = SE_R_ + SE_ABS_ + SE_MIR_ is true. SE_MIR_ can be ignored provided the studied sample is thinner than the wavelength of the incident radiation, or the measured SE_TOT_ is higher than 15 dB [20]. SE_TOT_ can be calculated as ${\mathrm{SE}}_{\mathrm{TOT}}=-10\log_{10}T$ and its reflection and absorption components, as ${\mathrm{SE}}_{R}=-10\log_{10}\left(1-R\right)$ and ${\mathrm{SE}}_{\mathrm{ABS}}=-10\log_{10}\left(T/1-R\right)$, respectively.

## 3. Results and Discussion

EMI shielding efficiency and its components are commonly used quantities to describe materials for potential shielding applications. By calculating the shielding efficiency and the analysis of the contribution of its reflection and absorption components, it is possible to distinguish the main EMI shielding mechanism. Figure 4 shows typical SE_TOT_ and SE_R_ of PLA/GNP composites with GNP of 25 μm lateral size in function of frequency in the microwave range from 0.2 to 12 GHz (see Appendix A for analogical graphs for composites with GNP of 0.2 and 5 μm lateral size). The spectra should be analyzed with respect to the reference sample—pure PLA marked with a black dashed line in the graph. SE_TOT_ shows an explicit dependency on the nanocarbon filler loading in the composite. The higher the GNP loading, the stronger the shielding properties. SE_TOT_ shows no clear dependence on the frequency, except for the reference sample and low GNP loading samples (under 5 wt%), which show a shielding growth at frequencies close to 12 GHz. That growth is reproduced in the reflection component at similar frequencies. Contrary to SE_TOT_, SE_R_ does not show dependency on the GNP content, except for the peak close to 12 GHz. Instead, the SE_R_ values of PLA/GNP composites are comparable to those of pure PLA.

To distinguish the main EMI shielding mechanism, we have investigated the SE_R_:SE_TOT_ ratio based on the full SE_TOT_ and SE_R_ spectra. The SE_TOT_ used for the calculations was calculated as the difference between the spectrum of a PLA/GNP composite and pure PLA. We have noticed that the average SE_R_:SE_TOT_ ratio decreases with the growing GNP loading. For composites with the highest GNP loading, the SE_R_ makes on average 3.5, 2.2, and 0.8% of total shielding efficiency for 0.2, 5, and 25 μm GNP lateral size, respectively. We conclude that the main shielding mechanism in PLA/GNP composites is absorption, and the graphene nanofiller is the source of absorption. Regarding SE_ABS_, as the SE_R_ is low, and the SE_ABS_ is the difference between SE_TOT_ and SE_R_, the absorption component follows the values of SE_TOT_.

In polymer composites, high EMI shielding is usually related to electrical conductivity and the dominant reflection component [26,27]. However, in our previous works, we have already shown a selection of polymer composites with absorption as the main EMI SE component in the terahertz range [28,29,30]. The main shielding mechanism can change with the nanocarbon loading in the composite. For example, Kashi et al. show a shift from the reflection mechanism to the absorption mechanism in PLA composites with 9 and 15 wt% GNP loading, respectively [31].

To better understand the influence of nanocarbon inclusions on the SE_TOT_, we have plotted the SE_TOT_ relative to the reference sample in the function of GNP loading at three selected microwave frequencies—2, 6, and 10 GHz (see Figure 5). Regardless of the frequency and size of GNP, the relation between SE_TOT_ and GNP loading is linear-like. The specific slope of the SE_TOT_(GNP wt%) changes with the size of graphene flakes and the frequency at which the shielding efficiency was measured. The rate at which the SE_TOT_ grows with the growing GNP loading decreases with the increasing frequency, e.g., from ~2 to ~1 dB/wt% for 2 and 10 GHz, respectively, in 0.2 μm composites. The highest growth rate is exhibited by the 25 um composites and equals ~3 dB/wt% at 2 GHz. The size of the GNP used influences the maximum SE_TOT_ exhibited by a sample. Figure 5 shows that the bigger the lateral size of GNP used, the higher the maximum relative SE_TOT_. A 25 um sample exhibited the highest relative SE_TOT_ of 42 dB with 15 wt% GNP loading at 2 GHz. At 6 GHz, the addition of 15 wt% of GNP enhances the SE_TOT_ in relation to the lowest GNP loading by ~19, ~62, and ~256 times, in the case of composites with the nanofiller particle size of 0.2, 5, and 25 μm respectively. That influence of the nanofiller shape and size on the absorbing properties of a composite has been noticed by Chamorro-Posada et al. in the terahertz range [21]. They have shown that composites imbued with nanofillers with higher aspect ratios exhibit a higher absorption coefficient, which is directly related to the EMI shielding properties of a material. Here, we show that also in the microwave range, polymer composites enhanced with particles with a higher aspect ratio show higher EMI shielding efficiency.

When analyzing Figure 5, it is clear that SE_TOT_ is not constant in the measured frequency range. SE_TOT_ values can vary by up to 8 dB, depending on the sample. Usually, the SE_TOT_ value decreases with growing frequency. However, the decrease of SE_TOT_ with increasing frequencies might not have any physical meaning and be caused by the technical imperfections of the measurement setup. We can postulate such an outcome, as the waveguide with an empty sample holder also shows the frequency-dependent decrease in SE_TOT_.

The influence of nanofiller geometry is also visible in the absorption coefficient shown in Figure 6a. The absorption coefficient is an important quality, especially in materials whose main shielding mechanism is absorption. The absorption coefficient is measured as α=Abs/d, where *d* is the thickness of a sample and *Abs* is the absorbance of the material, which is calculated as $Abs=-\ln\left(T/1-R\right)$. As seen in Figure 6a, the absorption coefficient changes linearly with the GNP loading, similar to the SE_TOT_. Like shielding efficiency, α depends on the size of GNP dispersed in the composite. The highest α is exhibited in composites with a GNP size of 25 μm and equals 67 cm^−1^ for a sample with 15 wt% GNP loading (measured at 5 GHz).

EMI shielding efficiency is often related to the conductivity σ of a polymer composite, as Shukla reports both SE_TOT_ components depend on the σ in the following way—SE_ABS_ ∝σdμ and SE_R_ ∝ σ/α [20]. Both EMI shielding efficiency and conductivity depend on forming so-called conductive paths within the insulating polymer matrix. Figure 6b shows conductivity in the function of GNP loading. The relation is not linear such as in the case of SE_TOT_ or α. Instead, especially in the case of composites with large GNP, the relation shows an exponential character. Interestingly, the conductivity values only show high diversity at 15 wt% loadings in regard to the average lateral size of a GNP. A sample with the largest GNP particles and 15 wt% GNP loading exhibited the highest measured conductivity of ~116 S/m, which is an exceptional result in the case of PLA/nanocarbon composites. For example, Spinelli et al. show similar PLA composites with GNP and MWCNT additions, which exhibit maximum DC conductivity of 6.27 and 4.54 S/m for composites with 12 wt% of GNP and MWCNT, respectively [26].

The absorption of EM radiation by a material often leads to heat generation within the material. To avoid the overheating of an electronic device, it is important to ensure good heat management. Heat management can be investigated by measuring thermal properties such as thermal conductivity or heat dissipation (Figure 7). Similar to EMI shielding efficiency and electrical conductivity, thermal conductivity κ grows with growing GNP content. The size of nanofillers influences thermal conductivity. The composites with the largest GNP lateral size showed much higher κ than those with smaller GNP lateral size. This might be caused by the more facile formation of conducting paths in the material with larger graphene flakes or the fact that heat is more easily transferred within a single graphene layer than between parallel layers forming a flake [19]. The highest measured κ 1.72 W/mK was exhibited by a composite loaded with 15 wt% of 25 μm GNP. This value is over five times higher than that of a pure PLA sample.

A similar study has been conducted by Kargar et al., which reported on EMI shielding efficiency and thermal conductivity of epoxy composites imbued with GNP of two different thicknesses—up to 3 and 12 nm [3]. Here we show that not only the thickness but also lateral size of a GNP is of utter importance to the potential heat management of material. Thermal conductivity measured for a PLA/GNP composite with 15 w% of 25 μm GNP is consistent with the values for epoxy/GNP composite reported by Kargar et al. [3], however, it is much higher than similar PLA based composite reported by Spinelli et al. [26]. Thermal conductivity shows exponential growth with GNP loading, similar to electrical conductivity. We assume this relation shows the common origin of electrical and thermal conductivity in polymer composites infused with conductive nanofillers, which we have previously stated as the formation of conductive paths. The relation between nanofiller loading and thermal conductivity in highly loaded composites has been investigated by Kargar et al. [16], showing similar results.

To put our work in perspective, we compare the results obtained for the PLA/GNP composite with the addition of 15 wt% of 25 μm GNP with the literature (see Table 1). In most cases, except for an epoxy/FLG composite reported by Kargar et al. [3], our composite shows the highest EMI shielding effectiveness while exhibiting over the average thermal conductivity, at a relatively low thickness (1 mm).

## 4. Conclusions

We report a study on physical properties—EMI shielding efficiency, absorption coefficient, the electrical and thermal conductivity of multifunctional PLA/GNP composites, and the influence of flake size on said properties. We have studied composites with three average lateral sizes—0.2, 5, and 25 μm at six GNP loadings (0.5 to 15 wt%). Regardless of the flake size, all composites showed good EMI shielding properties—over 10 dB at higher GNP loadings (>10 wt%). The composite exhibiting the highest EMI SE, σ, and κ was the composite with the addition of 25 μm GNP at the highest loading (15 wt%). The composite exhibited EMI SE of over 40 dB in the microwave range for a 1 mm thick sample, translating to an absorption coefficient of around 60 cm^−1^, σ of ~116 S/m at 5 GHz, and κ of 1.72 W/mK at room temperature. This study shows that introducing a grid of randomly distributed conducting flakes substantially enhances shielding, electric and thermal properties of a given polymer. We state that all physical properties studied here can be tuned by changing the nanofiller loading and its aspect ratio. The amelioration of α, σ and κ can be achieved by changing the filler loading or choosing a nanofiller with a different aspect ratio. However, the combination of the two yields the most effective results, as shown in this study. The composites studied in this work may be used as nonconductive dual-functional material at limited weight applications such as aerospace and mobile electronics.

## Figures and Tables

**Figure 1 materials-14-02856-f001:**
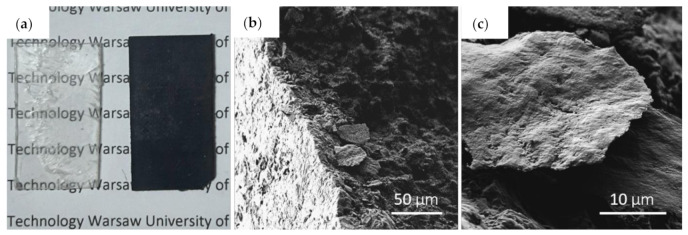
PLA/GNP composite. (**a**) Picture of a reference PLA sample (transparent) and a composite sample (black). SEM pictures of (**b**) cross-section of a PLA/GNP composite showing GNP dispersed in the PLA matrix and (**c**) close-up on a single GNP.

**Figure 2 materials-14-02856-f002:**
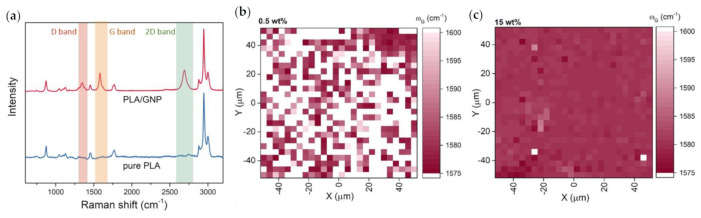
(**a**) Typical Raman spectra of a pure PLA sample and PLA/GNP composite with GNP characteristic bands highlighted. Raman maps showing G band position of (**b**) a sample with 0.5 wt% GNP and (**c**) a sample with 15 wt% GNP.

**Figure 3 materials-14-02856-f003:**
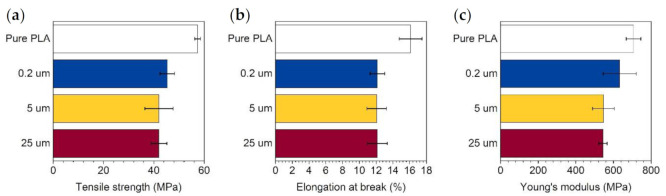
Tensile strength test results of testing species comprising 10 wt% of GNP of various flake size: (**a**) tensile strength results, (**b**) elongation at brake results and (**c**) Young’s modulus results.

**Figure 4 materials-14-02856-f004:**
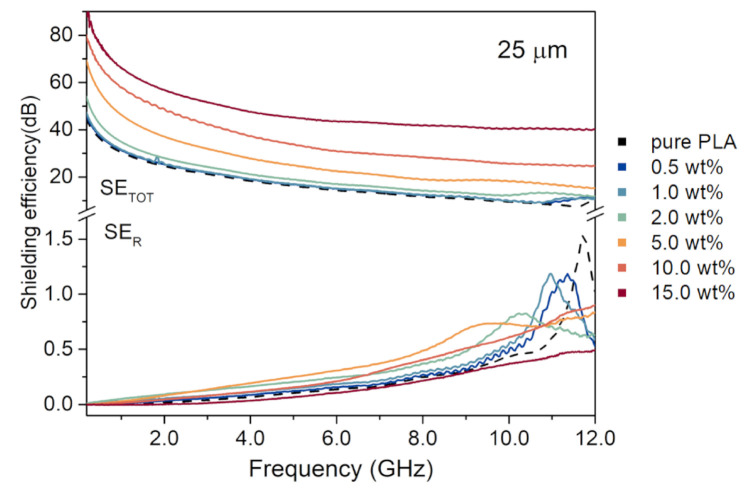
Raw data of total shielding efficiency and its reflection component of PLA/GNP composites with different GNP loading and 25 μm GNP.

**Figure 5 materials-14-02856-f005:**
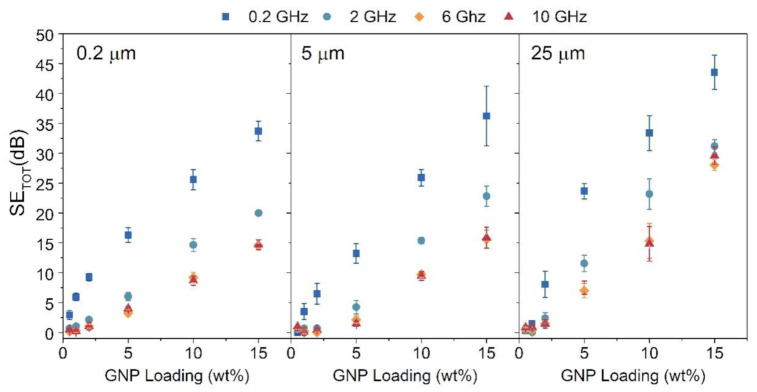
Total shielding efficiency of PLA/GNP composites at three different frequencies. Each panel shows data for different graphene flake size. Data is plotted in relation to GNP loading.

**Figure 6 materials-14-02856-f006:**
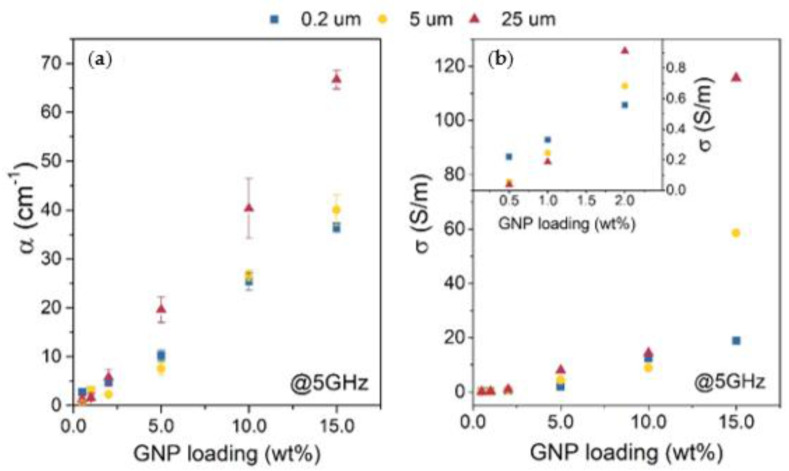
(**a**) Absorption coefficient in function of GNP loading at 5 GHz and (**b**) conductivity in function of GNP loading with an inset graph of low GNP loading composites. With common legend above the graphs.

**Figure 7 materials-14-02856-f007:**
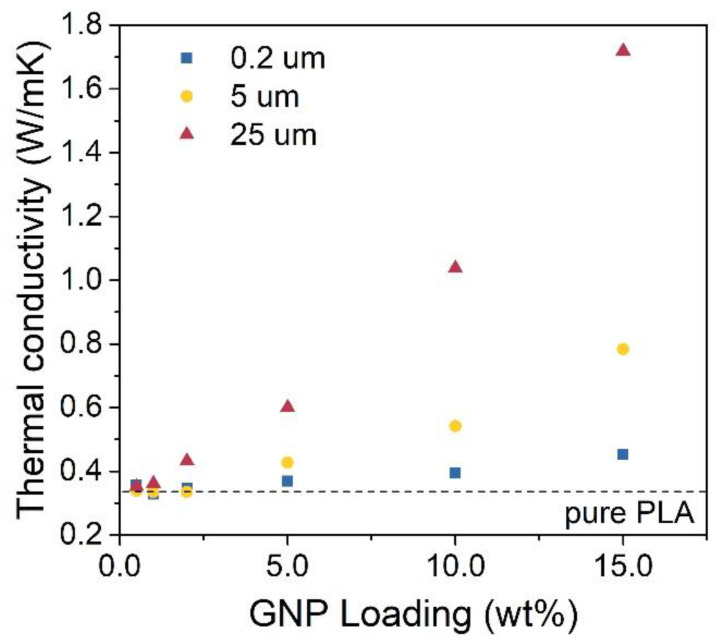
Thermal conductivity of PLA/GNP composites.

**Table 1 materials-14-02856-t001:** State of the art polymer composites with nanocarbon additives exhibiting high EMI SE or κ, together with our composite PLA/GNP with 15 wt% of 25 μm GNP.

Type of Composite	EMI SE (dB)	Range (GHz)	Thickness (mm)	κ (W/mK)	Ref.
PLA/GNP	16	8–12	1.2	-	[32]
PA6/GNP	16	8–12	1.0	-	[33]
SR/GNP	20	8–12	1.7	-	[34]
PMMA/GNP (foam)	13–19	8–12	4.0	-	[35]
PLA + PEO/GNP	14	8–12	1.5	-	[36]
Silicone/GNP	-	-	0.1–1.5	2.60	[37]
PPS/GNP	-	-	-	4.41	[18]
PA6/graphene-GO	-	-	-	2.14	[38]
GNP/epoxy	-	-	-	2.67	[39]
BE/graphene	-	-		0.54	[40]
PLA/GNP	10.22	30	10.0	0.66	[26]
Epoxy/FLG	45	8.2–12.4	1.0	8.00	[3]
PLA/GNP	42	0.2–12	1.0	1.72	

## Data Availability

The raw data required to reproduce these findings are available on request from the corresponding author, K.Z.-C. The processed data required to reproduce these findings are available on request from the corresponding author, K.Z.-C.

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
