# Peer review of "Graphene Infused Ecological Polymer Composites for Electromagnetic Interference Shielding and Heat Management Applications"

_materials, 2021, doi:10.3390/ma14112856_

Round 1

Reviewer 1 Report

Comments on Manuscript entitled Graphene infused ecological polymer composites for electro magnetic interference shielding and heat management applications 

  1. In the abstract the authors should include all the range of frequency where they report measurements, especially since in the 8-12 GHz they only show data at 10 GHz. Also, the author should avoid very general statements like those in lines 19-22 and better present information relevant for the specialist reading the paper
  2. The authors should underline in the introduction what was previously done in the field of composites and stress better the novelty of the present paper with respect to those of other authors, mentioned in the reference section but not only.
  3. SEM images were taken at very low magnification, being impossible to actually evidence  GNP; moreover, the dimension of GNP used for the presented is not included. The authors should add images of PLA and GNP separately, with higher magnification, and comment on the morphology. The authors should include SEM images of composites with various lateral size and illustrate and comment on the similarities and differencies in aspect and morphology. Also, they should include at least for one type of GNPs, images for various percentage of GNPS
  4. In Raman, the spectra of the 3 varieties of GNPs should be included, the authors should mention which was the type of GNP used in the presented data. They should add data with various concentration of GNP and various lateral size.
  5. Please comment on the lateral resolution of Raman microscopy with respect to the dimension of the GNP with lateral size of 0.2, 5 and 25 microns, used  in the composite materials.
  6. The authors should present  tensile strength , elongation at brake and Young’s modulus results for all the investigated compositions, e.g. for GNP concentration from 0.5, to 15 wt% (weight percent) of graphene in the mixture and comment on the obtained results. Also they should include in the manuscript information on the relevance of these tests for the desired application of the multifunctional composites.
  7. Please provide reference for the statement in lines 129-130 However, the tensile strength of obtained composites is higher than for neat aliphatic polyolefins, e.g. HDPE.
  8.  It is not clear why the authors chose to present sparce data under the characterization methods section. Please include in the manuscript all the relevant data for characterizing the produced samples.
  9. No data points presented in Figures 5, 6 and 7 have error bars. However, they should be included. As such, the authors should provide measurement errors, e.g to measure several times the same sample, and also sample errors, e.g measurements should be performed for samples obtained in the same conditions, in order to check on the  reproducibility. Please add such data in the graph and comment on the the errors.
  10. Please comment why most of the SE TOT values obtain with GNP of 5 microns are lower than that obtained with 0.2 microns.
  11. The authors mention that the decrease of SETOT with growing frequencies might not have any physical meaning and be caused by the technical imperfections of the measurement setup. This outline the significance of error bars. Moreover the authors should include the sketch of measuring system and the  measurements of SE TOT with the empty cell, as function of the frequency (in Figure 4). Also, if there is no physical meaning, it is not clear why the authors present all the data sets for 2, 6 and 10 GHz in Figure 5.
  12. In figure 6a the authors should include a legend. It is not clear why they chose to show results on 2,6 and 10 GHz in previous Figure and those on 5 GHz here. For Figure 6b), the authors mention that relation shows an exponential character - they should include the exponential fitting, the resulted constants of fitting, and they should explain this character.
  13. Authors should include always the units for the presented value (see for example 6.27 and 4.54- no units!
  14. Both electrical and thermal conductivity are related to GNP, therefore the authors should include values for the pure GNP materials, as function of their lateral size. 
  15. Thermal conductivity present as well an exponential behavior with the GNP content, please check, fit,  and provide explanation on this behavior.
  16. The authors should consider a real discussion section of the results combining the material characteristics such as composition, morphology, mechanical electrical and thermal properties to the functional properties.
  17. Please revise the conclusion section upon modifying the manuscript according tot the previous suggestions. Please avoid being too general in concluding section - avoid sentences like This study shows that introducing a grid of randomly distributed conducting flakes substantially enhances shielding, electric and thermal properties of a given polymer.

Reviewer 2 Report

This article reports on dual-functional po(lylactic acid) polymer composites infused with graphene nanoplatelet for electromagnetic interference shielding and heat management applications. There are some issues to be taken care of throughout the manuscript.

Here are my concerns:

  1. The novelty of this work is unclear. In particular, the authors should clarify the novelty of this work.
  2. The abstract should be more informative to present the findings of the work.
  3. The introduction should be a clear statement about what the challenge was and how the challenge was resolved.
  4. The authors should discuss and summarize the previous studies of graphene polymer composites for EMI shielding properties.
  5. How the authors optimize the composite wt% ratio? Why the authors is selected the 15 wt %? Please provide the composites dielectric properties results with different wt%.
  6. The authors should provide the performance comparison (thermal conductivity and shielding efficiency) with previously reported work.

Reviewer 3 Report

Please find comments in the attached pdf file.

Reviewer 4 Report

A study on EMI shielding efficiency, absorption coefficient, electrical and thermal conductivity of multifunctional PLA/GNP composites is presented and the role of inclusions-flake size and concentrations on these properties is experimetally investigated.

The results are very interesting for the scientific community. However, some revisions are suggested to the authors in order to support their findings.

In particular, in order to accomplish with standard measurements requirements, attention has to be given to:

  1. the sample dimensions - Please, specify the geometrical characteristics of the sample under test in the different experimental setup.
  2. the reproducibility of the results - Please, report some indication concerning the statistics of the experimental characterizations (mechanical/thermal and electrical), starting from the number of the sample under test.
  3. the used instrumentation: Please, complete the description of that

Moreover, as it concern the reported DC conductivity results (fig.6b), it could be opportune to specify how they are obtained. Are they measured? In which way? Furthermore, the low concentration region is not fully understandable being the data overlapped. An inset figure zooming this part is suggested.

Finally, by overall looking at the results, despite the higher dimension of the filler seem to improve the detected performance at the highest explored filler concentration, for 5% and/or 10% filler amount, a different behaviour appears (figs. 4-5-6-7). Have the authors some justification or interpretation of this?

Round 2

Reviewer 2 Report

The paper quality is improved after the revision.

Author Response

We thank Reviewer for this comment.

Reviewer 4 Report

The reviewed version of the original paper is in line with the suggested revisions.

Nevertheless, the new version suffers of problem with respect to the referenced papers list. Therefore it is suggest to check the references, with particular attention to the new inserted Table 1.

Author Response

We thank the Reviewer for pointing out the problem, we have fixed that in the new revised manuscript.